# Subjective Well-Being and Future Orientation of NEETs: Evidence from the Italian Sample of the European Social Survey

Cristiano Felaco [1] and Anna Parola [2,*]

1. Department of Social Sciences, University of Naples Federico II, Vico Monte della Pietà 1, 80138 Naples, Italy
2. Department of Humanities, University of Naples Federico II, Via Porta di Massa 1, 80133 Naples, Italy
* Correspondence: anna.parola@unina.it

**Abstract:** The acronym 'NEET' includes adolescents and young people aged 15–34 years not engaged in education, employment or training programs. According to recent studies, NEET represents a high-risk category to suffer from lower well-being and mental health problems. Following a life course approach, this study examined the self-reported subjective well-being and the future orientation of NEETs. To do this, the study used the latest European Social Survey data (Round 9—2018), limiting our analysis to Italian respondents aged 15–34 years. The final sample included 695 participants. Descriptive analysis and Student's t-test were performed to compare the subjective well-being and the future orientation of NEETs with those of non-NEET young adults. We hypothesize lower subjective well-beings in the NEET group and more difficulties in future planning than in the non-NEET group. Then, a mediation path model was carried out to study the relationship between employment condition (non-NEET/NEET) and subjective well-being through future orientation. The path model showed the mediator role of future orientation. Results indicated that future orientation plays a role in mitigating the effect of the unemployment condition on well-being. Starting from these findings, practical implications regarding career guidance interventions are discussed.

**Keywords:** NEET; young people; subjective well-being; future orientation; mediation analysis

## 1. Introduction

The term 'NEET' includes adolescents and young people not engaged in education, employment, or training programs. The acronym, which first emerged in the UK in the late 1980s, has become a broader and more exact measure of exclusion compared with the youth unemployment indicator (Eurofound 2016).

Although the age range defining NEET often varies among EU countries, Eurostat focuses on the population aged 15–34 years. In 2021, the lowest NEET rates are found in the Scandinavian countries and Northern Europe in general. Among member states of the European Union, Italy had the largest NEET population, where more than 20 percent of all young people are neither in employment nor in education or training (Eurostat 2022). The number of NEETs increases with age since most adolescents aged 15–19 years in the EU remain in education and training (6.8%). For the population aged 20–24 years, the NEET rate was 14.8%, while for those aged 25–29 years, the rate was 17.3%. The gender gap is still present in relation to presence in the labor market. A total of 14.5% of females aged 15–29 years were NEET, while the corresponding rate among males was 11.8%.

Addressing youth unemployment inevitably requires an examination of the financial and economic conditions of countries. The share of NEET youth followed the economic downturn of the late 2000s, and the current crisis related to the COVID-19 pandemic raises similar concerns. According to the International Labor Organization (2020), young people represent the most at-risk group to experience labor market exclusion due to the economic and financial consequences of COVID-19.

In this study, we examine the self-reported subjective well-beings and future orientations of NEET and non-NEET youth in Italy using the European Social Survey (ESS) data.

We frame our study from a life-course paradigm (Shanahan et al. 2016; Schoon and Bynner 2017, 2019). According to this approach, the school-to-work transition is an important developmental task for young people. Young people who remain out of the labor market and are not engaged in education or training programs represent a high-risk category to suffer from a lower well-being (Jongbloed and Giret 2022).

Quantitative and qualitative studies (Felaco and Parola 2020; Gaspani 2018; Parola et al. 2022) have shown how NEET individuals can experience difficulties in planning for the future. Moreover, a negative future time perspective can have effects on psychological health with both internalizing and externalizing problems (Parola et al. 2022).

The aim here is to investigate the relationship between employment condition in young adults (NEET and non-NEET) and subjective well-being, while analyzing the potential mediator role of future orientation. We hypothesized that NEETs are more likely to suffer from a lower subjective well-being and future orientation, and that future orientation might play a role in mediating this relationship.

## 2. Literature Review

### 2.1. NEET Phenomenon in a Life-Course Approach

Although the NEET rate for young people is closely linked to economic performance and the business cycle, NEETs are often examined without considering the socioeconomic context, and most studies fail to understand the highest rate in the EU South (Mascherini 2019; Kotroyannos et al. 2015; Zuccotti and O'Reilly 2019).

In a recent paper, Avagianou et al. (2022) proposed a theoretically informed empirical examination of the spatiotemporally uneven expansion of the NEET population between 2008 and 2018 in the EU South. As the authors said, "the weak integration of youth—particularly female—into many of the regional labour markets examined is related to spatially dependent structural barriers and institutional insufficiencies. It is also related to various cultural norms that affect the agency of young people" (Avagianou et al. 2022, p. 17). Three factors are proposed by the authors as possible explanations: (a) local economies' structural constraints; (b) formal institutions, more specifically, the skill-mix market demands and the quality of the educational system; and (c) family and social relations.

Guided by a life-course approach (Shanahan et al. 2016; Schoon and Bynner 2017), characteristics of institutional structures and transition systems have a key role in shaping the school-to-work transition and buffering the effects of economic recession. According to the life-course theory, school-to-work transitions can be defined as a developmental task for young people because it is a status transition in the institutionalized life course (Shanahan 2000). Young people are called to manage a sequence of status–role transitions and role configurations based on social structures. In Western societies, the transition from school to work has been delayed due to the fewer employment opportunities that changed following the economic recession. Therefore, NEET status seems to be moving along a trajectory of disadvantage (Bynner and Parsons 2002) that the economic recession reinforced (Schoon and Bynner 2019). This may be particularly true in countries with stagnating economies, such as Italy. Within the transition regime model, Italy was included in the sub-protective transition regime, which is characterized by mostly comprehensive education, less developed vocational training, and insecure employment and limited social protection of the labor market (Walther 2006; Schoon and Bynner 2019).

### 2.2. NEETs and Well-Being

Although at the social level, the NEET status represents a considerable loss in terms of unused productive capacity and a substantial welfare payment cost (Eurostat 2022), the young people pay the biggest costs. NEET represents a high-risk category to suffer from a lower well-being.

Subjective well-being is defined as people's appraisals and evaluations of their own lives (Diener et al. 2010) and includes reflective cognitive judgments, such as life satisfaction and emotional responses to ongoing life in terms of positive and negative emotions.

The literature on subjective well-being has grown enormously in the past decade (Diener et al. 2018). Several studies have underlined the internal and external determinants of subjective well-being (Schimmack 2006). Positive subjective well-being is associated with significant improvements in several outcomes such as health and longevity, supportive social relationships, citizenship, work performance, and resilience (for a review, see De Neve et al. 2013), as well as lifestyle in general (Nie et al. 2021; Zheng and Ma 2021). Conversely, poor health, separation, unemployment, and a lack of social contact are all strongly negatively associated with subjective well-being (Dolan et al. 2008).

According to the most recent review of subjective well-being (Diener et al. 2018), one of the more highly replicated findings in this area is that unemployment is strongly associated with lower subjective well-being.

A recent study by Jongbloed and Giret (2022) examined the effects of NEET status on subjective well-being across 24 EU countries. The authors found that NEET status has a strong, significant negative effect on well-being as compared to the non-NEET population. Moreover, several studies showed that the unemployment condition in youth is associated with distress (Bjarnason and Sigurdardottir 2003; Stea et al. 2019), depression (Crowe and Butterworth 2016; Bartelink et al. 2019), and anxiety (Virtanen et al. 2016; Bartelink et al. 2019). A recent systematic review (Gariépy et al. 2021) shed light on the directionality of these associations and highlighted that several longitudinal studies found that mental health problems at an early age increase the vulnerability to becoming NEET (Goldman-Mellor et al. 2016; López-López et al. 2019; O'Dea et al. 2016; Power et al. 2015; Rodwell et al. 2018), while only a few studies showed evidence for the inverse relationship (Gutierrez-Garcia et al. 2017). Moreover, NEET condition predicted later suicidal behavior (Gutierrez-Garcia et al. 2017) and alcohol use disorder (Gutierrez-Garcia et al. 2017; Manhica et al. 2019).

### 2.3. NEETs and Future Orientation

Future orientation is an individual's subjective view of the future (Seginer 2009). As a general concern for and corresponding consideration of one's future, it may shift depending on situational factors (Kooij et al. 2018). As such, future orientation is conceptualized as self-contextualizing (Zimbardo and Boyd 1999).

Several works studied how young people face the future (Brannen and Nilsen 2002; Devadason 2008). As Gaspani (2018) noted, studies confirmed the growing inability to plan long-term and the need to neutralize future uncertainty.

This "fear of the future" seems to be more felt in NEET individuals. Studies showed that the NEET condition leads young people to perceive the future as more difficult and complicated (Parola and Marcionetti 2022). According to Piumatti et al. (2014), the lack of career opportunities may increase the precariousness of young people's future career plans.

Moreover, the perception of uncertainty about the future may have negative psychological effects in terms of individuals' health (De Witte 2005; Sutin and Costa 2010). Indeed, future orientation is crucial for positive development outcomes (Nurmi 2005) and is crucial for well-being, motivation, and behavior (Kooij et al. 2018; Sun and Shek 2012).

Parola and Donsì (2019) compared the time perspective following Zimbardo and Boyd's (1999) paradigm of the fatalistic and hedonistic present and future orientations in students, employees, and NEET youths. The study showed that the NEET youths are present-fatalistic, focusing on the control of the events. Hence, NEET youths presented a lower level of future orientation than the students and employees.

### 2.4. Links between NEET Condition, Future Orientation, and Well-Being

Several studies showed the link between socioeconomic status (SES) and future orientation and highlighted that a higher SES is most likely associated with a future-oriented



time perspective (Fuchs 1982; Singh-Manoux and Marmot 2005). Guthrie et al. (2009) showed that a future-oriented time perspective seems to be typical of individuals with a high level of education and was more present in the sample of employees.

The perception of uncertainty about the future, in turn, may have negative psychological effects in terms of individuals' health (De Witte 2005; Sutin and Costa 2010). Positive future orientation is crucial for positive development outcomes (Nurmi 2005) and is crucial for well-being, motivation, and behavior (Kooij et al. 2018; Sun and Shek 2012). Prenda and Lachman (2001) showed that the anticipation of attaining value goals in the future operated as a protective factor on mental health. Moreover, as noted by Shipp et al. (2009), individuals with a lower level of future orientation have a more pessimistic view of their future that, in turn, leads to the worry of an unpredictable future and increased feelings of anxiety. In a recent meta-analysis, Kooij et al. (2018) reported that positive future orientation was associated with life satisfaction and subjective health but not with happiness. Moreover, individuals with a higher level of future orientation reported the highest levels of well-being (higher levels of life satisfaction and subjective health, and lower levels of anxiety and depression).

Previous studies have shown the mediator role of future orientation between SES and health (Guthrie et al. 2009), and the moderation role of future time perspective between employment condition and mental health such as anxiety, depression, withdrawal, and aggressive behavior (Parola and Marcionetti 2022). To our knowledge, no research has studied the mediation role of future orientation between employment condition (NEET vs. non-NEET) and the dimension of subjective well-being.

## 3. The Current Study

In this study, we examine the subjective well-being and future orientation of NEET and non-NEET youth in Italy using the European Social Survey (ESS) data. Moreover, we tested the mediator role of future orientation between employment condition and subjective well-being.

Most studies that have addressed the difficulties of school-to-work transitions, such as the transition to adulthood, have adopted the life-course paradigm (Buchmann and Kriesi 2011). This approach is based on the assumption that historical change influences individual development, and individual development in the aggregate can influence historical change (Johnson et al. 2011). Transition systems reflect the features of national institutional and structural arrangements, including the education system, the labor market, the welfare system, and family structures (Raffe 2008, 2014). The ESS adopts a full comparative perspective and provides a unique resource on the changing social, political, and economic conditions in Europe (Schnaudt et al. 2014). The protocol administered by different European countries allows for looking at the findings in the contexts of single-country as well as cross-national analyses.

Based on the above frameworks, we hypothesized that NEET youth are more likely to experience lower levels of subjective well-being (H1), and future orientation (H2). Moreover, we hypothesized a negative relationship between future orientation and subjective well-being (H3). Finally, we hypothesized that future orientation might play a role in mediating the relationship between employment condition and subjective well-being (H4). The conceptual model is shown in Figure 1.

To verify our hypothesis, we applied the mediation path analysis to data from the European Social Survey (ESS).

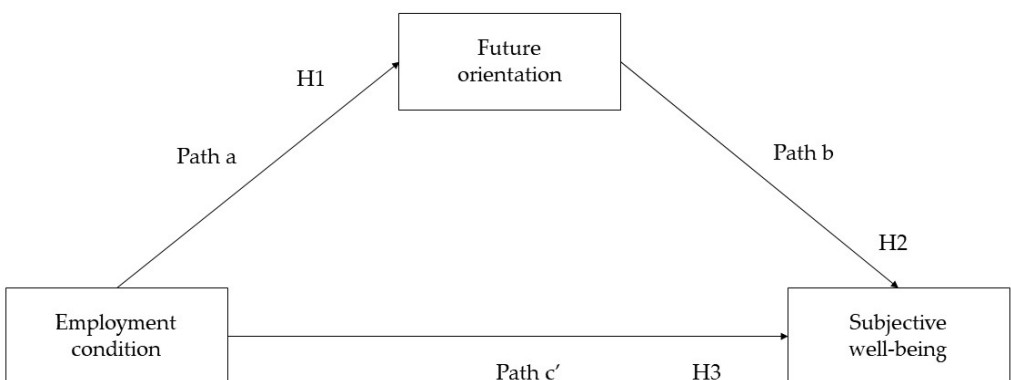

**Figure 1.** Statistical Diagram.

## 4. Materials and Methods

ESS is a large-scale and cross-national research program on basic human values, attitudes, beliefs, and behavioral patterns of different populations in more than thirty countries. We made use of the Round 9 data, which refers to the year 2018. At this moment, Round 9 refers to the latest data collected in Italy.

The population targeted by the ESS is all non-institutionalized individuals aged 15 years and over regardless of their legal status, nationality, and language, and a sampling strategy using a strict random probability design ensures the representativeness of the sample.

We limited our analysis to Italian young NEETs; to do so, we filtered the matrix such that we only had respondents aged 15–34 years who lived in Italy during the survey period. In our sample, NEETs are young people aged 15–34 years unemployed and not seeking a job.

To accomplish our research aims, we investigated the relationship between NEET status (non-NEET/NEET) and self-reported subjective well-being, also analyzing the potential mediator role of future orientation. Self-reported subjective well-being derived from index scores made by the combination of two variables selected from the ESS sample: self-reported life satisfaction (item used: "How satisfied are you with your life as a whole on a scale of zero to 10?") and levels of happiness (item used: "How happy are you on a scale of zero to 10?"). The two items used are in line with the theoretical concept of "subjective well-being" (Diener et al. 2010). Moreover, these two items have a Cronbach's alpha of 0.75.

The variable used as mediator was "Plan for future or take each day as it comes" which can be answered by participants between 0 (I plan for my future as much as possible) and 10 (I just take each day as it comes).

### Data Modeling

At first, we used the information obtained through the application of ESS Round 9 from 2018 to characterize the sample: gender, age, education, marital status, co-residence of parents and children, and typology of the place of residence. ANOVAs, LSD post-hoc tests, and linear regressions were performed to analyze the influence of sociodemographic variables on subjective well-being and future orientation.

To evaluate self-reported subjective well-being and future orientation, we performed mean, standard deviation, skewness, kurtosis descriptive statistical analyses, and a difference analysis (Student's t-test). Student's t-tests and bootstrap analysis (95% CI) estimated with 5000 bootstrap samples were performed to detect differences between the non-NEET group and the NEET group in well-being and future orientation. The effects of the differences were evaluated with Hedge's $g$.

The normal distribution of the sample was ensured by the values of skewness ($|Sk.| < 3$) and kurtosis ($|K.| < 10$).

Statistical analyses were performed with R software (R Core Team 2017) and the following packages: lavaan (Rosseel et al. 2015), corrplot (Wei et al. 2017), and graphViz via DiagrammeR (Iannone 2018). To test the hypothesis that future orientation mediates the relationship between employment status and subjective well-being, a mediation analysis was computed (Hayes 2017 with a 5000-bootstrap resampling procedure. The mediation analysis tested whether the indirect effect of the future orientation mediated the effect of employment status and subjective well-being with the bootstrapping confidence interval.

Considering the score distribution of the measured variables, the maximum likelihood (ML) estimator was used to conduct the following statistical analyses. In line with previous studies (Rossi et al. 2021), a 2-step approach was followed (Hayes 2017; MacKinnon 2012).

STEP 1: A predictor-only model was specified: the 'employment condition' (X) was regressed on 'subjective well-being' (Y).

STEP 2: the full mediation model was specified: the 'employment condition' (X) was regressed on 'subjective well-being' (Y) through 'future orientation' (see Figure 1).

All the reported regression coefficients were unstandardized (β).

## 5. Results

The characteristics of the sample are presented in Table 1. The sample comprised 695 participants who were almost equally distributed by gender, mostly unmarried because we only considered individuals up to 34 years old (mean age about 25 years), and with a mean of 12.4 years of education. Moreover, just over a fifth of the sample was composed of young NEETs, and the main source of income was salary or wages for little more than 40 percent of participants, whereas more than half of them had no personal income. Most of the sample lived in the countryside.

**Table 1.** Sociodemographic characteristics of the sample (N = 695).

| Variable | Attributes | n (%)/M (SD) | Sk./K. |
|---|---|---|---|
| Gender | Female | 343 (50.6) | |
| | Male | 352 (49.4) | |
| Marital status | Married/civil union | 103 (14.9) | |
| | Separated/divorced | 9 (1.3) | |
| | None of these | 579 (83.8) | |
| Years of education | | 13.20 (3.383) | 0.314/0.118 |
| Age | | 24.72 (5.931) | 0.105/−1.156 |
| Employment condition | Non-NEET | 153 (22.2) | |
| | NEET | 535 (77.8) | |
| Income source | Employment | 276 (43.6) | |
| | Others | 19 (3.7) | |
| | None | 319 (52.7) | |
| Residence area | Large cities and surroundings | 129 (18.6) | |
| | Medium cities and towns | 237 (34.2) | |
| | Countryside | 326 (47.1) | |

M = mean; SD = standard deviation; Sk. = skewness; K. = kurtosis.

ANOVAs revealed a significant effect on the following variables: well-being per employment conditions $F(1, 685) = 25.503$, $p < 0.001$, $\eta^2 = 0.036$); respondent's income source $F(2, 601) = 4.107$, $p < 0.05$, $\eta^2 = 0.036$); future orientation per employment conditions $F(1, 686) = 15.915$, $p < 0.001$, $\eta^2 = 0.023$); and the respondent's income source $F(2, 602) = 8.846$, $p < 0.001$, $\eta^2 = 0.013$). Linear regression showed that future planning was predicted by years of education (β = −0.248, $t = -6.623$, $p < 0.001$) and age (β = 0.155, $t = -4.068$, $p < 0.001$); in addition, well-being was predicted by years of education (β = 0.102, $t = 2.666$, $p < 0.01$) and age (β = −0.172, $t = -4.585$, $p < 0.001$).

Student's t-tests are reported in Table 2. Statistically significant differences between the non-NEET group and the NEET group were found. The non-NEET group reported a higher mean than the NEET group for the well-being dimension ($t = 5.050$, $p < 0.001$). Moreover, the non-NEET group had a lower mean than the NEET group for the future orientation variable. In other words, these findings indicate that non-NEETs show a higher positive subjective well-being and tend to plan more for the future.

**Table 2.** Descriptive analyses, *t*-test, and effect sizes for NEET group and non-NEET group.

| | Non-NEET | NEET | *t*-Test | | | |
| | M (SD) [95% CI] | M (SD) [95% CI] | T(df) | 95% CI | g | Sk./K. |
|---|---|---|---|---|---|---|
| Well-being | 7.496 (1.400) [7.376,7.610] | 6.778 (1.992) [6.458,7.082] | 5.050 * (685) | [0.439,0.998] | 0.417 | −1.027/ 1.585 |
| Future orientation | 4.58 (2.696) [4.36,4.81] | 5.58 (2.939) [5.12,6.05] | −3.975 * (685) | [0.379,1.058] | 0.465 | 0.214/ −0.737 |

M = mean; SD = standard deviation; 95% CI = confidence interval at 95%; *df* = degrees of freedom; g = Hedge's *g*; * $p < 0.001$; Sk. = skewness; K. = kurtosis.

A mediation path analysis was conducted to grasp the effect of employment condition on subjective well-being and the role of future orientation in the relationship.

The mediation path analyzed whether employment condition influenced subjective well-being and whether future orientation mediated the effect of employment condition on subjective well-being (Figure 1).

This research employed the bootstrapping mediation test for the analysis, which provided the indirect effect.

The direct effect (Figure 2 and Table 3) indicates that young people in NEET condition are likely to have a lower subjective well-being ($\beta = -0.665$, SE = 0.143, $p < 0.001$).

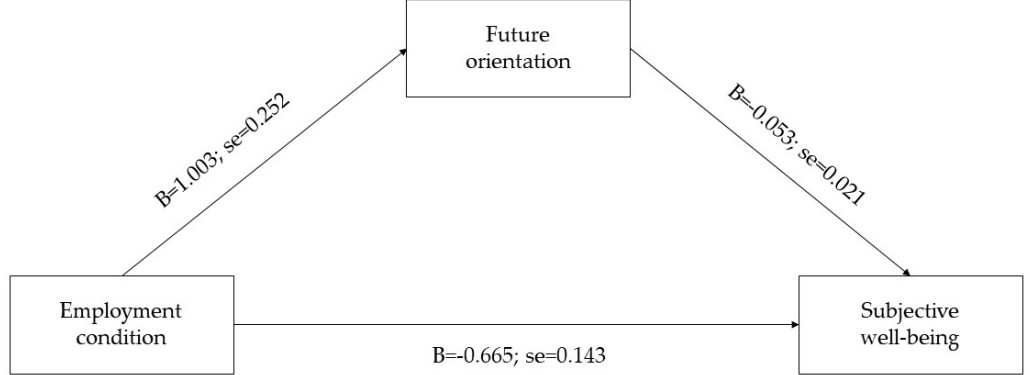

**Figure 2.** Mediation model.

**Table 3.** Direct and indirect effects of employment condition on subjective well-being via future orientation.

| | B | se | 95% CI[L-U] |
|---|---|---|---|
| Path a | 1.003 | 0.252 | [0.508; 1.495] |
| Path b | −0.053 | 0.021 | [−0.095; −0.011] |
| Path c′ | −0.665 | 0.143 | [−0.947; −0.384] |
| Indirect effect | −0.053 | 0.029 | [−0.116; −0.005] |

B = unstandardized regression coefficient, se = standard error; 95% CI = confidence interval at 95%.

Moreover, individuals who experience the NEET condition are likely have a lower future orientation ($\beta = 1.003$, SE = 0.252, $p < 0.001$).

Finally, individuals that have a low future orientation are likely to have a lower subjective well-being ($\beta = -0.053$, SE = 0.021, $p < 0.05$).

Bootstrapping analysis indicates that the indirect effect was significant ($\beta = -0.053$, SE = 0.029, $p < 0.01$, CI [$-0.116$, $-0.005$]).

## 6. Discussion

The NEET acronym refers to young people not engaged in education, employment, or training. Although the phenomenon of unemployment has always been a social problem, in recent years, the interest in the psychological effects of youth unemployment has been increasing.

This may be particularly true because the current world in which young people are called to find a job has different characteristics compared to the past. Among the major threats and challenges young people are facing, there are socioeconomic policies and inequalities and digitalization. In terms of socioeconomic policies and inequalities, the labor market is characterized by job insecurity, temporary assignments, and fragmented career paths that make the transition to the world of work more difficult. In terms of digitalization, the technological evolution has, in some cases, contributed to the worsening of working conditions and increased wage disparities (Ford 2015), along with racial (Kang et al. 2016) and gender (Dastin 2018) discrimination.

Moreover, the COVID-19 pandemic also has a not negligible impact because it has exposed and exacerbated existing inequities in the labor market (International Labor Organization 2020).

In this scenario, youth represent the most vulnerable category. As reported by Blustein et al. (2020), the growing precarity of work is having particularly painful impacts on young people across the globe.

Guided by a life-course approach (Shanahan et al. 2016; Parola et al. 2022), the school-to-work transition consists of a crucial developmental task. The non-adaptive school-to-work transition (Parola et al. 2022) has devastating effects on the psychological health and well-being of individuals (Bartelink et al. 2019).

In this study, we focused on the effects of the NEET condition on subjective well-being and future orientation and the mediation role of future orientation between employment condition and subjective well-being. To this end, we have used the European Social Survey data (Round 9) related to the Italian context.

First, in a preliminary descriptive analysis, we briefly compared the subjective well-being and future orientation among NEET and non-NEET groups. Results showed lower levels of perceived well-being in the NEET subsample than in non-NEET group. Moreover, the subsample of NEET youths presented a tendency to live more on the day-to-day than the non-NEET group.

Then, a mediation path model was performed to analyze whether the employment condition influenced subjective well-being while the future orientation mediated the effect of employment condition on subjective well-being.

According to our H1, we confirmed the relationship between employment condition and subjective well-being. Young people of NEET status are likely to have a lower subjective well-being. This evidence was in line with previous studies that highlighted the effect of unemployment condition on subjective well-being (Jongbloed and Giret 2022). A recent Italian longitudinal study (Bonanomi and Rosina 2022) showed that passing from a non-NEET to NEET status leads to a lower level of well-being, while the exit from NEET status leads to significant increases in well-being.

In line with H2, we confirmed the relationship between employment condition and future orientation. Individuals who experience the NEET condition have a lower tendency to plan for the future. The weak future orientation is related to the idea of a bleak (Bynner and Parsons 2002) or dark future (Zaleski et al. 2019). According to Bynner and Parsons

(2002), for individuals who are not engaged in education, employment, or training, the future may be bleak. They tend not to worry about the future because their mind is occupied with what is happening in the present moment (Sobol-Kwapinska et al. 2016).

Moreover, we also confirmed the relationship between subjective well-being and future orientation (H3). According to several studies (Kooij et al. 2018; Mohammed and Marhefka 2020), among the time perspective dimensions, difficulty in future planning has profound effects on different negative outcomes, such as well-being.

Finally, the mediation test supported the proposed H4 by showing the mediator role of the future orientation. In other words, future orientation may foster the relationship between employment condition and subjective well-being. Future orientation theories emphasize the motivational importance of future goals by linking individuals' present behaviors to their future goals (Hawkins and Blakeslee 2004).

These findings may contribute to stressing the negative outcomes of youth unemployment in the Italian context. As we said above, Italy reported the highest number of individuals in the NEET condition (Eurostat 2022). Such a condition in Italy is also fueled by the problematic school-to-work transition. In recent years, psychological and sociological studies have studied the Italian school-to-work transition compared with other European countries (Brunetti and Corsini 2019; Pastore et al. 2022). In particular, Pastore (2019) highlighted that the slowness of school-to-work transition is a consequence of the "sclerotic" labor market with a very low job-finding rate and an education system that is not fully efficient. The consequence of the Italian education system is that most young Italians find a permanent job in their thirties.

While we will not go into detail on the analysis of EU policies, our study suggested the importance of supporting the individual in school-to-work transitions because the unemployment condition affects their well-being.

*Limitations and Future Research Directions*

This study was not free of limitations. Firstly, the ESS datasets have a correlational nature, which does not allow conclusions about causal relationships between variables. A longitudinal panel design would be desirable to understand the direction of effects. Secondly, we relied solely on self-report measures. We must consider that the data may be influenced by a reporting bias (e.g., social desirability). Finally, the limitation of using secondary data must be acknowledged. Pre-existing data are often limited in terms of the psychological variables considered. In this study, future orientation was assessed by a single item. Nevertheless, consistent with the recommendation of Weston et al. (2019) on the use of secondary data in psychology, using pre-existing data appears to be a powerful and low-cost tool for exploring crucial research questions. Moreover, in our study, thanks to the secondary data of the European Social Survey, we have had a great advantage in terms of accessing data on hard-to-reach populations, such as NEET youths.

**7. Conclusions**

The mediation analyses showed that future orientation plays an important role in the psychological process that leads to mitigating the negative effects between unemployment condition and well-being.

This study highlighted the need for career guidance interventions that allow one to plan and deal with the future. It is important to enhance beliefs about future outcomes and positive feelings about pursuing their goals and wishes (Fusco et al. 2022; Ginevra et al. 2016). This way, young people will be more able to imagine multiple possible future scenarios. Career guidance interventions should be oriented to help young people cope with labor market challenges, capturing the specific experience of failure in SWT and the related negative outcomes. Practices guided by the Life Design paradigm (Savickas 2012) appear to be the most useful for mastering career transition (Hartung 2019). LD intervention promotes the freedom of the individual to think about their future (Fusco et al. 2021) and find their way to build a prosperous and sustainable life (De Vos et al. 2020).

In conclusion, the present study could contribute to the ongoing debate concerning the NEET condition, fostering the development of outreach strategies for young NEETs aimed to identify and reach them, intercept their needs, and then make tailored services.

**Author Contributions:** Conceptualization, A.P.; methodology, C.F.; data curation, C.F; formal analysis, A.P. and C.F.; writing, A.P. and C.F. All authors have read and agreed to the published version of the manuscript.

**Funding:** This research received no external funding.

**Conflicts of Interest:** The authors declare no conflict of interest.

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
