# Peer review of "Subjective Well-Being and Future Orientation of NEETs: Evidence from the Italian Sample of the European Social Survey"

_socsci, doi:10.3390/socsci11100482_

Round 1

Reviewer 1 Report

The article analyzes is very interesting and important practical and research problem.

In order for an article to meet scientific requirements, all of the comments below must be addressed:

1.       Introduction - The authors discuss the initially observed problems in theory and practice. Please add the aim of article in this section. Moreover the research questions must be formulated.

2. This article does not have a literature review section. Maybe it's worth adding and expanding literature about the theory and researching other authors?

3.       Definitely before explaining your methodology (point 2), emphasize what is the research gap?

4.       I propose to write not only discussion, but also conclusions and future research directions.

5. The literature review is too small

Author Response

Before we outlined in detail the changes we have made to the manuscript, we would like to thank you for your invaluable feedback. We are well aware of the time and effort you have invested in the review process and we have benefited greatly from your comments and suggestions. We have tried to address all of the concerns raised and—as you will see below—have followed the advice given in most cases. We hope you share our impression that the quality of the manuscript has improved substantially.

Following a point-by-point response:

  1. Introduction - The authors discuss the initially observed problems in theory and practice. Please add the aim of article in this section. Moreover the research questions must be formulated.

Thanks for your comment. We have added the aim in the introduction section and formulated the research questions (in yellow in the text).

  1. This article does not have a literature review section. Maybe it's worth adding and expanding literature about the theory and researching other authors?

Thanks for your comment. We have added a separate section (see 2. Literature review) and a new subsection (see 2.4 Links between NEET condition, future orientation, and well-being)

  1. Definitely before explaining your methodology (point 2), emphasize what is the research gap?

Thanks for your comment. We have emphasized the research gap in the section 2.4.

  1. I propose to write not only discussion, but also conclusions and future research directions.

Your advice is very enlightening and we are very grateful. We have added a separate section for limitations and future research directions, and conclusion.

  1. The literature review is too small

Thanks for your comment. We have integrated the literature review in the introduction section (in yellow in the text).

Reviewer 2 Report

This is an interesting and thought provoking paper, key terms such as NEET and self defined well being are considered and adequately referenced.

It is well known that being NEET is related to feelings of lower self-worth and well being, the key contribution here is the link between the Italian economic and socio-political context and the findings presented here, therefore more context around Italy and the importance of this country in relation to findings would be most welcome and would allow the author(s) to better scope their contribution to knowledge.

It would also be beneficial to learn more about the life-course approach and how this research method 'fits' with the wider survey methodological design. A clear definition outlining the life-course approach and its links to the broader methodological design would help improve the overall coherence of the methodology section.

Author Response

Before we outline in detail the changes we have made to the manuscript, we would like to thank you for your invaluable feedback. We are well aware of the time and effort you have invested in the review process and we have benefited greatly from your comments and suggestions. We have tried to address all of the concerns raised and—as you will see below—have followed the advice given in most cases. We hope you share our impression that the quality of the manuscript has improved substantially.

Following a point-by-point response:

  1. This is an interesting and thought provoking paper, key terms such as NEET and self defined well being are considered and adequately referenced. It is well known that being NEET is related to feelings of lower self-worth and well being, the key contribution here is the link between the Italian economic and socio-political context and the findings presented here, therefore more context around Italy and the importance of this country in relation to findings would be most welcome and would allow the author(s) to better scope their contribution to knowledge.

Thanks for your comment. We have illustrated the results in light of the Italian economic and socio-political context (in yellow in the text).

It would also be beneficial to learn more about the life-course approach and how this research method 'fits' with the wider survey methodological design. A clear definition outlining the life-course approach and its links to the broader methodological design would help improve the overall coherence of the methodology section.

Thanks for your comment. We have discussed the life-course approach and its links to the broader methodological design in the methodology section (in yellow in the text).

Reviewer 3 Report

In the introduction section I suggest to provide a previous studies related subjective well-being in general, then you can highlight the research gap, you can pointe out the gap by “NEET” issue. finally, the research contribution need to be included in this section.

You can check the following literature: 

Nie, P., Ma, W., & Sousa-Poza, A. (2021). The relationship between smartphone use and subjective well-being in rural China. Electronic Commerce Research, 21(4), 983-1009.

Zheng, H., & Ma, W. (2021). Click it and buy happiness: does online shopping improve subjective well-being of rural residents in China?. Applied Economics, 53(36), 4192-4206.

Zheng, H., & Ma, W. (2022). Scan the QR Code of Happiness: Can Mobile Payment Adoption Make People Happier?. Applied Research in Quality of Life, 1-12.

Rahman, M. S., Andriatmoko, N. D., Saeri, M., Subagio, H., Malik, A., Triastono, J., ... & Yusuf, Y. (2022). Climate disasters and subjective well-being among urban and rural residents in Indonesia. Sustainability, 14(6), 3383.

Wijayanto, H. W., Lo, K. A., Toiba, H., & Rahman, M. S. (2022). Does Agroforestry Adoption Affect Subjective Well-Being? Empirical Evidence from Smallholder Farmers in East Java, Indonesia. Sustainability, 14(16), 10382.

In the data analysis the author only employed descriptive statistics and t-test. to improve the valuable of this study I suggest to add some econometric approach that can provide empirical analysis about the influence of sociodemographic in table 1 on subjective well-being.

Author Response

Before we outline in detail the changes we have made to the manuscript, we would like to thank you for your invaluable feedback. We are well aware of the time and effort you have invested in the review process and we have benefited greatly from your comments and suggestions. We have tried to address all of the concerns raised and—as you will see below—have followed the advice given in most cases. We hope you share our impression that the quality of the manuscript has improved substantially.

Following a point-by-point response:

  1. In the introduction section I suggest to provide a previous studies related subjective well-being in general, then you can highlight the research gap, you can pointe out the gap by “NEET” issue. finally, the research contribution need to be included in this section.

You can check the following literature: 

Nie, P., Ma, W., & Sousa-Poza, A. (2021). The relationship between smartphone use and subjective well-being in rural China. Electronic Commerce Research, 21(4), 983-1009.

Zheng, H., & Ma, W. (2021). Click it and buy happiness: does online shopping improve subjective well-being of rural residents in China?. Applied Economics, 53(36), 4192-4206.

Zheng, H., & Ma, W. (2022). Scan the QR Code of Happiness: Can Mobile Payment Adoption Make People Happier?. Applied Research in Quality of Life, 1-12.

Rahman, M. S., Andriatmoko, N. D., Saeri, M., Subagio, H., Malik, A., Triastono, J., ... & Yusuf, Y. (2022). Climate disasters and subjective well-being among urban and rural residents in Indonesia. Sustainability, 14(6), 3383.

Wijayanto, H. W., Lo, K. A., Toiba, H., & Rahman, M. S. (2022). Does Agroforestry Adoption Affect Subjective Well-Being? Empirical Evidence from Smallholder Farmers in East Java, Indonesia. Sustainability, 14(16), 10382.

Thanks for your comment. We have added the aim in the introduction section and formulated the research questions. Moreover, in the literature review we have added more general studies of well-being following your suggestions (in yellow in the text).

In the data analysis the author only employed descriptive statistics and t-test. to improve the valuable of this study I suggest to add some econometric approach that can provide empirical analysis about the influence of sociodemographic in table 1 on subjective well-being.

We have performed the required analysis (see data analysis and results sections). Thank you for the comments!

Round 2

Reviewer 1 Report

Thank you very much for improving the article.

The article deals with an interesting subject and meets the substantive requirements.

The article should be published.